# Defining the Impact of Social Drivers on Health Outcomes for People with Inherited Bleeding Disorders

**DOI:** 10.3390/jcm11154443

**Published:** 2022-07-30

**Authors:** Karina Lopez, Keri Norris, Marci Hardy, Leonard A. Valentino

**Affiliations:** 1National Hemophilia Foundation, New York, NY 10001, USA; knorris@hemophilia.org (K.N.); mhardy@hemophilia.org (M.H.); whybloodclots@gmail.com (L.A.V.); 2Rush University, Chicago, IL 60612, USA

**Keywords:** inherited bleeding disorders, social determinants of health, social drivers of health, quality of life, bleeding, pain

## Abstract

The ways in which the social drivers of health, also known as the social determinants of health (SDOH), affect health outcomes for people with inherited bleeding disorders (PwIBDs) is unclear. This systematic review of the published literature examines the impact of SDOH on health outcomes in PwIBDs. Articles that included the following parameters in PubMed informed this study: published in English between 2011–2021; available in free full text; study population diagnosed with an inherited bleeding disorder; and study measured at least one of the clinical/non-clinical outcome measures: bleeding frequency, chronic pain, mortality, quality of life (QOL), and/or cost. The main findings from the 13 included articles emphasized the unmet need for reducing the economic burden with sustainable population health strategies and treatment options for PwIBDs. Rural location was also a significant contributor to both delayed diagnosis and decreased access to care. Furthermore, the need for a multidisciplinary comprehensive care team to address physical, psychosocial, and emotional needs of PwIBDs was raised as a priority target in the desire for equitable and optimal health. This systematic literature review suggests that the SDOH are associated with inferior health outcomes and may influence the clinical progression of inherited bleeding disorders.

## 1. Introduction

As defined by Healthy People 2030, “social determinants of health (SDOH) are the conditions in the environments where people are born, live, learn, work, play, worship, and age that affect a wide range of health functioning, and quality of life (QOL) outcomes [1]”. We feel these constructs are actually Social Drivers of Health rather than Social Determinants of Health because “determinants” suggests nothing can be done to change our health fate [2]. As such, we refer to the SDOH as social drivers of health interchangeably with more common social determinants of health phrasing. The SDOH include non-medical factors like socioeconomic status, education, neighborhood and physical environment, employment, social support networks, and access to quality health care [3] (Figure 1).

According to the World Health Organization (WHO), addressing the SDOH appropriately is fundamental for improving health and reducing long-standing inequities in health [4]. As outlined in Figure 1 and Figure 2, each framework organizes the key components into categories, establishing broad areas across multiple domains into one framework. While the literature surrounding the influence of the SDOH is robust among chronic conditions such as cancer, diabetes, and cardiovascular disease, the ways in which the SDOH affect health outcomes for people with inherited bleeding disorders (PwIBDs) remains unclear [5,6,7].

The inherited bleeding disorders community includes more than 3 million Americans [8]. The most frequently occurring bleeding disorders include von Willebrand Disease (VWD), hemophilia A, and hemophilia B. Less frequent, yet also serious, inherited bleeding disorders include deficiencies of factors II, V, VII, XI, XIII, fibrinogen, α2- antiplasmin, and platelet disorders such as Gray platelet syndrome and Glanzmann thrombasthenia [8]. These bleeding disorders prevent the blood from clotting normally, thus causing affected individuals to experience prolonged bleeding after an injury, surgery, or physical trauma [8]. While there have been great advances in the development of products to treat some bleeding disorders, this population faces a variety of difficult challenges which create barriers to care and potential disparities for health outcomes.

Health inequities for PwIBDs may be due to geographical location, financial barriers, access to quality care and treatment, educational hurdles, and psychosocial obstacles [9,10,11,12]. For many PwIBDs, the distance to their hemophilia treatment centers (HTCs), specialized, multidisciplinary health-care centers providing team-based care to address physical, psychosocial, and emotional needs, can affect how often they see their health care provider, if ever [5]. Some PwIBDs experience financial barriers related to cost of clotting factor products, insurance coverage/caps, copay assistance program issues, and out-of-pocket costs [5,10,13,14]. PwIBDs may experience bias in treatment, such as non-white PwIBDs reporting higher levels of chronic pain, thereby affecting their overall quality of care [5,9,12]. There is a lack of awareness among this chronic disease population and their medical providers regarding the signs of a bleeding disorder, leading to delays in diagnosis and excess morbidity, as well as the importance of early therapy [9,14]. If left unaddressed, intervention strategies to reduce these barriers/disparities will continue to be disease-specific, often targeting individual and health systems’ factors without addressing the SDOH [11]. 

The apparent lack of literature investigating the impact of SDOH on health outcomes in PwIBDs led us to conduct this systematic literature review. To examine this, we focused on PwIBDs and disease progression, rather than people at risk of a bleeding disorder, and we examined multiple clinical and non-clinical outcomes including bleeding frequency, chronic pain, cost, mortality, and QOL. The purpose of the review is to examine the impact of a broad range of social drivers, rather than limit the review to a specific subset. Given the current literature and the diverse topics, the KFF adapted framework shown in Figure 2 was used to select variables for analysis and categorize current knowledge on the impact of the SDOH outcomes for PwIBDs.

## 2. Materials and Methods

### 2.1. Eligibility Criteria

The following inclusion criteria were used to determine eligible study characteristics: (a) published in English during the last decade (2011–2021), (b) published in PubMed, (c) available in free full text, (d) study population was diagnosed with an inherited bleeding disorder, (e) the study measured one the following clinical outcomes: bleeding frequency, chronic pain, mortality, QOL, and/or cost of treatment.

### 2.2. Information Sources and Search Strategy

A reproducible strategy was used to identify studies investigating the impact of the SDOH outcomes for PwIBDs following the Preferred Reporting Items for Systematic Reviews and Meta-Analyses (PRISMA) guidelines for literature reviews (Appendix A) [15]. The Medical Subject Headings (MeSH) heading for SDOH was not introduced until 2014 [16], therefore, we searched for articles three years prior to ensure inclusivity. A full description of the search terms and search process is illustrated in the appendix (Table 1).

### 2.3. Selection Process

Titles were reviewed to ensure the study targeted PwIBDs, and abstracts read to ensure inclusion of a SDOH and at least one outcome measure of interest including: 

Clinical/non-clinical health outcome measures:**Bleeding Frequency**: The main clinical manifestation of a bleeding disorder is an increased bleeding tendency, either spontaneous or related injury, surgery, or physical trauma [8].**Chronic Pain**: Usually caused by arthritis in joints, a consequence of bleeds that have damaged the joint’s cartilage [12].**Mortality**: Inheritable bleeding disorders can lead to fatal complications if left untreated. Early detection and treatment can impact a person’s quality of life, thus impacting the SDOH [8].**Cost**: Considering that the total annual cost of treatment for many people with hemophilia is more than $250,000 per adult patient in the United States [16], cost can be an important consideration for the impact of the SDOH on PwIBDs.**Quality of Life**: The standard of health, comfort, and happiness experienced by PwIBDs, usually assessed through validated instruments/tools such as HR-QOL [8]. The amount of hemoglobin is a well-established measure of general health status for PwIBDs [16].

SDOH categories:**Health Care System**: access to healthcare and its quality (i.e., health insurance coverage and quality of care) [3].**Economic Stability**: finances (i.e., employment and income) [3]**Neighborhood/physical environment:** housing, environment (i.e., transportation, walkability, and safety) [3].**Community/social context:** ways a person lives, works, plays, and learns (i.e., social integration, support systems, and community engagement [3].**Education:** access to education and its quality (i.e., literacy and language) [3].**Food:** access to healthy/nutritious food (i.e., hunger) [3].

### 2.4. Data Collection Process and Study Risk of Bias Assessment

Data collection from the eligible articles is shown in Table 2, Table 3, Table 4, Table 5 and Table 6, where each table is specific to a SDOH category. The data was extracted for each article on the study design, study objective, number of participants, study population, and impact of outcome measured (Table 2, Table 3, Table 4, Table 5 and Table 6). The risk of bias and demonstration of causal relationship due to most of the articles being cross-sectional studies is discussed within the limitations section.

## 3. Results

### 3.1. Study Selection

The search resulted in 1466 original articles, and the titles reviewed for inheritable bleeding disorder populations resulted in 27 articles for retrieval (see the PRISMA flow diagram, Figure 3). Then, reviewing the titles for inherited bleeding disorder populations, SDOH, and language yielded 13 articles that were included in the systematic review (Appendix A). Several of the articles assessed multiple of the SDOH categories as health barriers, therefore there were overlaps of articles within each area. Overall, three were categorized as Economic Stability, one was categorized as Neighborhood and Physical Environment, zero were categorized as Food, two were categorized as Education, twelve were categorized as Community and Social Context, and four were categorized as Health Care Systems.

See Table 1, line 14 in the appendix for the search terms that generated the identified records.

### 3.2. Study Characteristics and Outcomes of Studies

Study designs included cross-sectional, case studies, and literature reviews. Sample sizes ranged from 30 to 798. Sample population and setting both varied, as well as the study results (Table 2, Table 3, Table 4, Table 5 and Table 6). Given that statistical analysis was not conducted in each of the studies, we reported on the main findings of the studies relative to the respective SDOH category and health outcome measured. Relative confidence intervals were included, if applicable. Table 7 incorporates all the SDOH components and respective clinical outcome. Each of the thirteen articles measured one or more of the clinical and/or non-clinical outcomes.

The articles that assessed Economic Stability analyzed barriers such as employment, financial income, and cost of treatment. The article by Burke et al., reported that economic instability was associated with considerable clinical, humanistic, and economic burden of hemophilia B in the U.S. [14]. The mean annual bleed rate was 1.73 (standard deviation, 1.39); approximately 9% of patients experienced a bleed-related hospitalization during the 12-month study period. Nearly all patients (85%) reported chronic pain, and the mean EQ-5D-5L utility value was 0.76 (0.24) [14]. Pinto et al., reported a significant impact of hemophilia on professional and economic levels where among the adult participants who had an occupation, either a full or a part-time job, or a student status, 28 (42.4%) reported missed days from work or school due to hemophilia (M = 28.57; SD = 55.67) [19]. Additionally, pain showed a wide duration range, varying from 1 month in three age groups to 612 months (51 years) in the adult’s group [18]. While Okide et al., conducted a literature review, a finding from published literature was that many community-dwelling adults with hemophilia may choose to work in jobs that are unsuitable for them to obtain or maintain insurance coverage [17]. At the same time, many with insurance coverage face rising costs of co-payments and lifetime restrictions [17].

Arya et al., was the only study to assess the SDOH category of Neighborhood and Physical Environment. The authors conducted the first study to assess healthcare providers’ perceptions around inequities in care for patients with inherited bleeding disorders living in Canada [5]. A main finding from this study was that rural location was felt to be a significant contributor to both delayed diagnosis and decreased access to care, the effects of which may not necessarily be mitigated by universal healthcare delivery. With regards to how long they estimated their patients’ travel time to clinic appointments to be, approximately half (48%, N = 34) estimated 30 min to one hour; 33% (N = 23) indicated one hour to two hours, 11% (N = 8) estimated over two hours, and a minority (6%, N = 4) said it took less than 30 min. [5]. 

Major themes analyzed for the articles that fell under Education were increased school absences and positive impacts of a prophylactic treatment. In the literature review from Okide et al., the authors recommend that community-based adult education and nursing interventions rooted in psychosocial counseling and support services can be helpful in this regard [17]. As reported by Pinto et al., among adults, work/school absences were reported by 42.4% participants and could last up to 293 days. These findings explicitly illustrate the negative impact of hemophilia on work- or school-related activities, corroborating the data from other surveys [18]. Nonetheless, PwIBDs who are on a prophylactic treatment have reported an effective reduction in the number of school and/or work absences [18].

The SDOH category of Community and Social Context analyzed common themes such as QOL and the benefits of sports participation within this population of PwIBDs. Atiq et al., reported that patients who participated in sports had a higher bleeding score item for muscle hematoma 0.38 ± 0.95 vs 0.24 ± 0.75 (*p* = 0.023), which remained significant after correction for age and sex (OR = 1.26, 95% CI: 1.04–1.53) [19]. Additionally, the authors found a linear association between more hours of physical activity per week and a better general health status (*p* < 0.001) [19]. Carroll et al., conducted a series of questionnaires, including the EQ-5D-3L and SF-36 version 2, where participants who reported a higher frequency of joint pain and history of joint surgery had statistically significantly lower EQ-5D-3L utility values than participants who did not experience joint pain or who had not had joint surgery previously (*p* < 0.001) [20]. In a cohort study, Gilbert et al., reported that hemophilia A carriers had significantly lower scores in the “Pain” and “General Health” domains of the Rand 36-item Health Survey 1.0 than controls [21]. Hemophilia A carriers had a median “Pain” score of 73.75 compared to a median score of 90 for control subjects (*p* = 0.02). For the “General health” domain, hemophilia A carriers had a median score of 75 compared to a median score of 85 for controls (*p*= 0.01) [21]. In one study by Goto et al., physical activity was reported to be positively correlated with bleeding risks among patients with severe and moderate hemophilia, where 55% of people with hemophilia who actively engaged in sports reported bleeding episodes associated with physical activity [22]. This study was the only study to touch on mortality out of the 13 articles, where the authors reported that physical inactivity is the fourth leading risk factor for mortality, accounting for 6% of deaths globally [22].

Articles pertaining to the Health Care System category analyzed themes such as access to care for minorities and for patients with rare bleeding disorders, geography, and gender bias. Arya et al., asked practitioners about access to care for visible minorities, patients of lower socioeconomic status (SES), for patients living in rural Canada. Specifically, when asked whether they believe that patients with low SES experience less access to care, 46% (N = 32) indicated ‘yes’; 39% (N = 27) indicated ‘no’, [5]. Moreover, participants felt that patients with bleeding disorders of an unknown cause receive less access to care as compared to those with a bleeding disorder of known cause a lack of certainty [5]. Forty-two percent of respondents (N = 29) felt that women with inherited bleeding disorders, including symptomatic carriers, experience less access to care as compared to men [5].

## 4. Discussion

This systematic review is the first to review, analyze, and synthesize literature regarding the impact of the SDOH outcomes in the inherited bleeding disorders population. Using a reproducible strategy, 1466 articles were identified, which were reviewed based on predetermined inclusion criteria. When categorizing by the KFF adapted framework, studies tended to cluster into the Community and Social Context, Economic Stability, and Health Care Systems categories. Few studies investigated the SDOH category of Education, and none of the studies investigated the category of Food. Most of the studies were cross-sectional, thus limiting the conclusions that can be made regarding causation. Nonetheless, based on the studies reviewed, an association was identified with bleeding frequency, chronic pain, cost, and QOL. This suggests that the SDOH have an association with inferior health outcomes and may influence the clinical progression of inherited bleeding disorders. The impact on mortality was seldomly measured, but when QOL was explored, the study findings did show an impact.

### 4.1. Summary of Evidence by SDOH

Articles categorized as Economic Stability considered topics related to employment, expenses, and cost of treatment. Burke et al., found a persistent and comprehensive economic burden of hemophilia B on patients receiving FIX prophylaxis, with substantial FIX treatment-driven costs to payers and society [14]. The direct non-medical and indirect costs associated with hemophilia B may comprise a relatively small proportion of the total cost, but nonetheless represent a significant burden to patients, employers, and society in the form of lost income and productivity for both patients and caregivers [14]. Direct non-medical costs were mainly driven by caregiver expenses, both professional and informal, and indirect costs were comprised largely of hemophilia-related unemployment and early retirement [14]. Furthermore, a survey conducted by Pinto et al., in Portugal showed that people with hemophilia do not experience higher unemployment than the general male population, nonetheless, 50% of the sample population declared an impact of the disease on their professional activity [18]. A literature review conducted by Okide et al., found that many community-dwelling adults with hemophilia may choose to work in jobs that are unsuitable for them to obtain or maintain insurance coverage [17]. Such a finding emphasizes the remaining unmet needs for reducing the economic burden with sustainable population health strategies and treatment options for people living with inherited bleeding disorders.

In the Neighborhood and Physical Environment category, the main finding was that rural location was felt to be a significant contributor to both delayed diagnosis and decreased access to care. These results are supported by qualitative interviews with PwIBDs where geographical barriers are a recurrent theme [5]. While there is no existing literature on specific geographic barriers for PwIBDs, the literature supports the concept of travel distance to their specialist being a potential barrier to care for persons with hemophilia [5]. The Hemophilia Experiences, Results, and Opportunities (HERO) initiative found that a significant number of young adults experienced difficulties visiting their HTCs with travel distance and travel time being the main barriers to accessing care. This has also been speculated to be the case for adults with hemophilia [5]. The study authors recommended telehealth or e-health initiatives to help minimize limitations to care for patients living in rural locations without easy physical access to an HTC.

The COVID-19 pandemic gave the opportunity for HTCs to further evaluate and identify new ways to address access and utilize digital innovation through the implementation of telemedicine. Ultimately, the need for social distancing while still delivering integrated, comprehensive care opened the ability to serve individuals who had previously faced challenges accessing an HTC. The relaxation of regulations enacted during the pandemic provided room to utilize telemedicine for HTC services, and broadening the HTC patient cohort whose geographic location was a significant barrier to accessing an HTC in some areas. While this is not a complete solution to pervasive health and social inequities, as not all individuals have access to a device and/or reliable internet, it was a positive step forward.

Few studies investigated the impact that a bleeding disorder has on Education. Pinto et al., referenced a study done by Shapiro et al., where it was observed that children with higher bleeding rates missed more days of school and tended to have lower academic achievement as compared to the general population [19]. Among adults, work/school absences could last up to 293 days [18]. Interestingly, the number of participants with a college education in Pinto et al.’s study was similar to that of the Portuguese general population in the 25–64 age group, thus showing that the impact of hemophilia is not reflected on lower educational achievement [18]. While there is a lack of evidence to show the impact of a bleeding disorder on educational status, research has shown that prophylactic treatment can effectively reduce the number of school and work absences [18], which is a great accomplishment for this patient population. 

The category of Community and Social Context mainly explored QOL outcome measure. Many of the articles focused around the area of physical activity. While sports participation and physical activity are associated with a better QOL and reduced frequency of spontaneous bleeding due to improved strength and fitness partners, many people with bleeding disorders lack physical activity due to their physical condition and fear of hurting themselves [19,22,23,24]. This indicates the need for strategies focused on the encouragement of regular engagement in physical activity among people living with bleeding disorders, promoting education about its benefits and potential risks, as well as guidelines on how to adequately deal with eventual injuries. 

Furthermore, the need for a multidisciplinary comprehensive care team to address physical, psychosocial, and emotional needs of people with hemophilia and other inherited bleeding disorders as a priority target in the improvement of health status and QOL in people with bleeding disorders was raised [14,17,18,19,27]. Given the COVID-19 pandemic, this is a crucial time to lend a voice to the importance of patient access to a multidisciplinary team to deliver whole-person, integrated, accessible, and equitable healthcare by interprofessional teams [28]. We call upon the medical community to lobby payors to continue to support integrated and accessible care for our patient communities. 

In the Health Care System category, a main finding by Arya et al., highlights that gender and geography are two key SDOH in the care for patients with inherited bleeding disorders. Travel distance and lack of healthcare provider awareness were noted to be potential barriers to care for patients with undiagnosed bleeding disorders and women alike [5,23,25]. Moreover, the study highlighted that factors outside of the SDOH, such as the degree and type of symptomatology, may contribute to diagnostic delay [5].

### 4.2. Summary of Evidence by Clinical/Non-Clinical Health Outcome

Bleeding frequency and chronic pain were the most measured clinical outcomes, with half of the articles finding statistical significance. Pain experience in hemophilia has been associated with poor psychological functioning, functional disability, and diminished QOL, additionally increasing the burden of the disease itself [17,20,21]. This association is an important finding of the review. In addition, Burke et al. found that despite individuals diagnosed with hemophilia B receiving FIX prophylaxis treatment, they continue to experience breakthrough bleeding and can eventually develop hemophilic arthropathy [14]. Mortality was seldomly measured, only being identified in one of the thirteen articles which was itself a literature review, so thus statistical significance did not apply. The article measuring this outcome was categorized in Social and Community Context. This suggests that more research is needed regarding the impact of the SDOH on mortality.

In terms of the two additional measured non-clinical outcomes, QOL was measured in twelve of the thirteen articles, with half of the articles finding statistical significance. Cost was rarely measured in two of the thirteen articles, with one of the two articles finding statistical significance. The studies measuring QOL were mostly located in the Social and Community category, whereas studies measuring cost were located amongst the Economic Stability and Health Care Systems categories.

## 5. Limitations

There are limitations to this study that are worth noting. The search was limited to free full text English articles published in PubMed in the last decade (2011–2021). Given that we did not receive any funding for this review, only free full text articles were reviewed. Since studies with significant results are more likely to be published, the studies in this review may reflect publication bias. The SDOH category of food was not explored in the literature, and thus only five of the six categories in the KFF adapted framework were analyzed. Additionally, the health outcomes were not defined and/or measured consistently throughout the different articles, hindering us from cross-comparing the outcomes effectively. Lastly, most articles were observational designs, impeding the ability to comment on causation. Conclusions from this review are therefore qualitative and meant to guide future research, which is why the National Hemophilia Foundation (NHF) will further explore this topic and make future analysis/recommendations.

## 6. Conclusions

This systematic literature review suggests that the SDOH has an association with health outcome and may influence the clinical progression of inherited bleeding disorders. Nonetheless, further research is needed to better characterize the direct impact of the SDOH outcomes in inherited bleeding disorders. Researchers should be careful to clearly define the health determinant of interested and ensure that the measure is deliberately being examined in studies. The impact of the SDOH key areas of neighborhood environments and food must be explored further, as they are likely contributors to bleeding disorder-related outcomes, excess morbidity, and excess cost of care for people living with bleeding disorders. Additionally, the SDOH literature must be strengthened, allowing for evidence-based interventions which will be useful in developing national policies for inherited bleeding disorders research and clinical care. Naming, defining, and understanding the root causes for the SDOH is the first step towards the mitigation of pervasive health and social inequities within the inherited bleeding disorders community to improve population health, reduce health disparities, and enable PwIBDs to thrive.

## 7. Implications

As representatives of the NHF, we would like to call to attention that for the past 40 years, HTCs have developed and perfected the multidisciplinary shared decision-making model [29]. We believe that those in our community who receive care at HTCs have greater likelihood of achieving the outcomes we desire for equitable and optimal health—longer lifespan, improved functioning, decrease in emergency room visits, hospitalization, and healthcare costs. The guidelines by which HTCs operate were established by the Medical and Scientific Advisory Council (MASAC) of NHF and the World Federation of Hemophilia’s Guidelines for Management of Hemophilia.

We are proud to report that HTCs are fulfilling the promise of high-quality primary care, as described in the article “In Implementing High-Quality Primary Care: A Report from the National Academies of Sciences, Engineering, and Medicine”, featured in JAMA’s May 2021 issue [30]. We continue to put improving health equity, access, and inclusion in our strategic plans for ongoing improvement to reach more individuals and remove barriers to care. Additionally, the COVID-19 pandemic facilitated the ability to serve PwIBDs who had previously faced challenges accessing an HTC through the implementation of telemedicine. While this is not a complete solution to pervasive health and social inequities, as not all individuals have access to a device and/or reliable internet, it was a positive step forward. Further, we propose that the comprehensive care delivered in HTCs can serve as the model for other chronic illnesses and medical conditions, including fostering open communication and collaboration with multidisciplinary teams, including primary care providers.

## Figures and Tables

**Figure 1 jcm-11-04443-f001:**
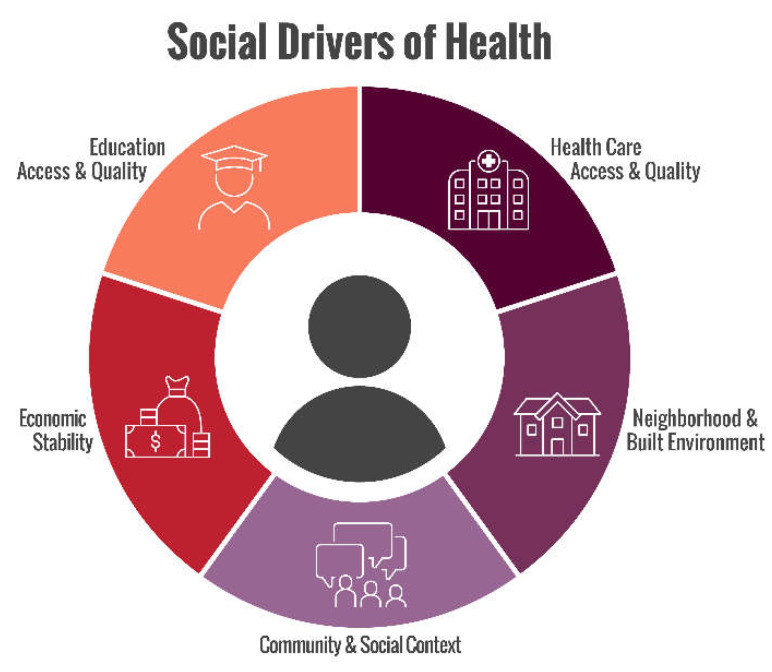
Social Drivers of Health framework adapted from Healthy People 2030.

**Figure 2 jcm-11-04443-f002:**
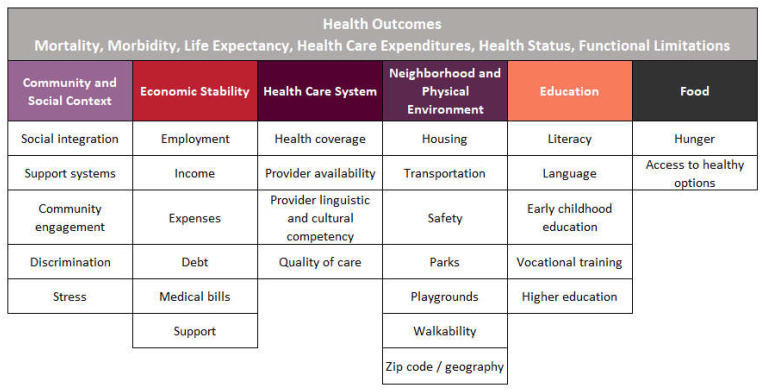
Social drivers of health framework adapted from Kaiser Family Foundation (KFF).

**Figure 3 jcm-11-04443-f003:**
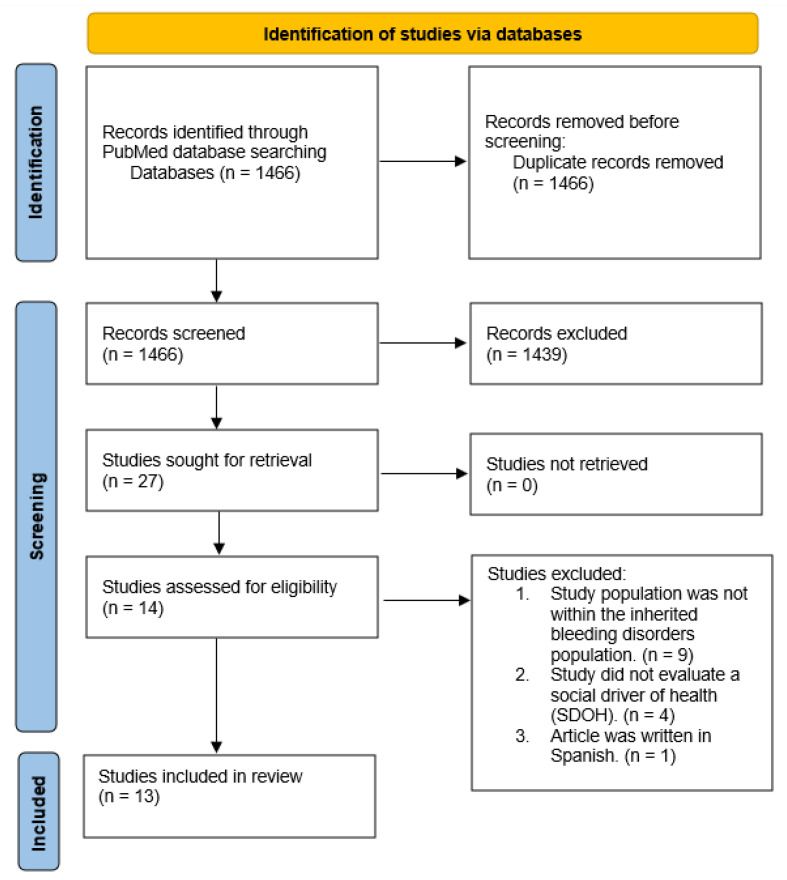
PRISMA flow diagram.

**Table 1 jcm-11-04443-t001:** Structure of Search and Search Terms.

Search Number	Search Terms	Number of Articles Retrieved
1	“Blood Coagulation Disorders” [Mesh] OR “inherited bleeding disorder*” [tw] OR IBDs [tw], OR “bleeding disorder*” [tw]	6441
2	“Social Determinants of Health” [Mesh] OR “Social determinants of health*” [tw] OR SDOH [tw] OR “social determinant of health” [tw]	10,114
3	#1 AND access healthcare AND avail * healthcare AND access care AND avail * care	12
4	#1 AND “Geography” [Mesh] or “zip code*” [tw]	3292
5	#1 AND “Literacy” [Mesh]	0
6	#1 AND “ESL” [tw] AND “English Second Language*” [tw]	0
7	#1 AND “Cultural Competency” [Mesh] OR “cultural competence*” [tw] OR “cultural sensitivity *” [tw]	3956
8	#1 AND “Social Support” [Mesh]	11
9	#1 AND “Quality of Life” [Mesh] OR “QOL *” [tw]	44,222
10	#1 AND “Food Insecurity” [Mesh] OR food insecurity *	5706
11	#1 AND “Health Personnel” [Mesh] AND “quality”	18
12	#1 AND #2	10,114
13	#1 AND #2-#12	6441
14	#1 AND #13 Filters: Free full text, from 2011 - 2021	1466

* Truncation: a PubMed search method in which symbols are used in place of letters or words to help you broaden your search.

**Table 2 jcm-11-04443-t002:** Summary of articles found focused on Economic Stability.

Author/Year	Title	Study Design	Objective	Number of Participants	Sample Population	Result
Burke et al., 2021 [14]	Clinical, humanistic, and economic burden of severe hemophilia B in the United States: results from the CHESS US and CHESS US+ population surveys	Cross-sectional	Used the CHESS US and CHESS US+ data to analyze the clinical, humanistic, and economic burden of hemophilia B for patients treated with factor IX prophylaxis between 2017 and 2019 in the US.	44 patient records in CHESS US and 57 patient records in CHESS US+	Hemophilia B patients	Nearly all patients (85%) reported chronic pain, and the mean EQ-5D-5L utility value was 0.76 (0.24). The mean annual direct medical cost was $614,886, driven by factor IX treatment (mean annual cost, $611,971). Subgroup analyses showed mean annual costs of $397,491 and $788,491 for standard and extended half-life factor IX treatment, respectively. The mean annual non-medical direct costs and indirect costs of hemophilia B were $2371 and $6931.
Okide et al., 2020 [17]	Challenges facing community-dwelling adults with hemophilia: Implications for community-based adult education and nursing	Literature review	Discussed the challenges facing community-dwelling adults with hemophilia and the implications for community-based adult education and nursing.	Final number of articles reviewed were not disclosed	Community dwelling adults with hemophilia	Sustainable efforts are needed in the provision of local, national and international leadership and educational resources to improve and sustain health care for community-dwelling adults with hemophilia.
Pinto et al., 2018 [18]	Sociodemographic, Clinical, and Psychosocial Characteristics of People with Hemophilia in Portugal: Findings from the First National Survey	Cross-sectional	Describe sociodemographic, clinical, and psychosocial characteristics of PWH of all ages in Portugal	146 males with hemophilia	106 adults, 21 children/teenagers between 10 and 17 years, 11 children between 6 and 9 years, and 8 children between 1 and 5 years	High prevalence of joint deterioration and pain, with the ankles and the knees being the most affected joints among all age groups. A significant impact of hemophilia on professional and economic levels was particularly evident. Moreover, significant anxiety and depression symptoms were found on 36.7% and 27.2% of adults, respectively, and a belief of chronicity and symptoms unpredictability was particularly prominent among adults and children/teenagers. QoL was moderately affected among adults, but less affected in teenagers and children.

**Table 3 jcm-11-04443-t003:** Summary of articles found focused on Neighborhood and Physical Environment.

Author/Year	Title	Study Design	Objective	Number of Participants	Sample Population	Impact on Outcome
Arya et al., 2020 [5]	Healthcare provider perspectives on inequities in access to care for patients with inherited bleeding disorders	Cross-sectional	Understand healthcare provider perspectives regarding access to care and diagnostic delay amongst this patient population.	70 respondents	Healthcare providers	Rural location was felt to be a significant contributor to both delayed diagnosis and decreased access to care. Such results are supported by our group’s presently ongoing qualitative interviews with patients with bleeding disorders, in which geographical barriers are a recurrent theme.

**Table 4 jcm-11-04443-t004:** Summary of articles found focused on Education.

Author/Year	Title	Study Design	SDOH Evaluated	Objective	Number of Participants	Sample Population	Result
Okide et al., 2020 [17]	Challenges facing community-dwelling adults with hemophilia: Implications for community-based adult education and nursing	Literature review	1.Education 2. Health care system 3. Economic stability 4. Community, Safety, and Social Context	Discussed the challenges facing community-dwelling adults with hemophilia and the implications for community-based adult education and nursing.	Final number of articles reviewed were not disclosed	Community dwelling adults with hemophilia	Sustainable efforts are needed in the provision of local, national and international leadership and educational resources to improve and sustain health care for community-dwelling adults with hemophilia.
Pinto et al., 2018 [18]	Sociodemographic, Clinical, and Psychosocial Characteristics of People with Hemophilia in Portugal: Findings from the First National Survey	Cross-sectional	1. Neighborhood and Physical Environment 2. Community, Safety, and Social Context 3. Education 4. Health care System 5. Economic Stability	Describe sociodemographic, clinical, and psychosocial characteristics of PWH of all ages in Portugal	146 males with hemophilia	106 adults, 21 children/teenagers between 10 and 17 years, 11 children between 6 and 9 years, and 8 children between 1 and 5 years	High prevalence of joint deterioration and pain, with the ankles and the knees being the most affected joints among all age groups. A significant impact of hemophilia on professional and economic levels was particularly evident.Moreover, significant anxiety and depression symptoms were found on 36.7% and 27.2% of adults, respectively, and a belief of chronicity and symptoms unpredictability was particularly prominent among adults and children/teenagers. QoL was moderately affected among adults, but less affected in teenagers and children.

**Table 5 jcm-11-04443-t005:** Summary of articles found focused on Community and Social Context.

Author/Year	Title	Study Design	Objective	Number of Participants	Sample Population	Result
Atiq et al., 2018 [19]	Sports participation and physical activity in patients with von Willebrand disease	Cross-sectional	Assessed the sports participation and physical activity of a large cohort of VWD patients.	798 VWD patients	474 had type 1, 301 type 2 and 23 type 3 VWD. The mean age was 39 ± 20 (standard deviation) years	Type 3 VWD patients more often did not participate in sports due to fear of bleeding and physical impair- ment, respectively, OR = 13.24 (95% CI: 2.45–71.53) and OR = 5.90 (95% CI: 1.77–19.72). Patients who did not participate in sports due to physical impairment had a higher bleeding score item for joint bleeds 1.0 (±1.6) vs 0.5 (±1.1) (*p* = 0.036). Patients with type 3 VWD and patients with a higher bleeding score frequently had severe limitations during daily activities, respectively, OR = 9.84 (95% CI: 2.83–34.24) and OR = 1.08 (95% CI: 1.04–1.12).
Okide et al., 2020 [17]	Challenges facing community-dwelling adults with hemophilia: Implications for community-based adult education and nursing	Literature review	Discussed the challenges facing community-dwelling adults with hemophilia and the implications for community-based adult education and nursing.	Final number of articles reviewed were not disclosed	Community dwelling adults with hemophilia	Sustainable efforts are needed in the provision of local, national and international leadership and educational resources to improve and sustain health care for community-dwelling adults with hemophilia.
Pinto et al., 2018 [18]	Sociodemographic, Clinical, and Psychosocial Characteristics of People with Hemophilia in Portugal: Findings from the First National Survey	Cross-sectional	Describe sociodemographic, clinical, and psychosocial characteristics of PWH of all ages in Portugal	146 males with hemophilia	106 adults, 21 children/teenagers between 10 and 17 years, 11 children between 6 and 9 years, and 8 children between 1 and 5 years	High prevalence of joint deterioration and pain, with the ankles and the knees being the most affected joints among all age groups. A significant impact of hemophilia on professional and economic levels was particularly evident. Moreover, significant anxiety and depression symptoms were found on 36.7% and 27.2% of adults, respectively, and a belief of chronicity and symptoms unpredictability was particularly prominent among adults and children/teenagers. QoL was moderately affected among adults, but less affected in teenagers and children.
Burke et al., 2021 [14]	Clinical, humanistic, and economic burden of severe hemophilia B in the United States: results from the CHESS US and CHESS US+ population surveys	Cross-sectional	Used the CHESS US and CHESS US+ data to analyze the clinical, humanistic, and economic burden of hemophilia B for patients treated with factor IX prophylaxis between 2017 and 2019 in the US.	44 patient records in CHESS US and 57 patient records in CHESS US+	Hemophilia B patients	Nearly all patients (85%) reported chronic pain, and the mean EQ-5D-5L utility value was 0.76 (0.24). The mean annual direct medical cost was $614,886, driven by factor IX treatment (mean annual cost, $611,971). Subgroup analyses showed mean annual costs of $397,491 and $788,491 for standard and extended half-life factor IX treatment, respectively. The mean annual non-medical direct costs and indirect costs of hemophilia B were $2371 and $6931.
Carroll et al., 2019 [20]	Real-world utilities and health-related quality-oflife data in hemophilia patients in France and the United Kingdom	Cross-sectional	Collect health-related quality-of-life (HRQoL) and health-utility data from hemophilia patients with differing disease severity	122 patients in France and 62 in the UK = 184 participants	Patients with hemophilia aged ≥12 years	The collected utility values reflected real-world data and can potentially serve as health-state weights in future cost–utility analyses, although it is important not to use EQ-5D-3L-, EQ-5D-5L-, and SF-6D-derived utility values interchangeably. The HRQoL data further documented the physical burden linked to hemophilia and its complications.
Gilbert et al., 2015 [21]	Haemophilia A Carriers Experience Reduced Health-Related Quality of Life	Cross-sectional	Test the hypothesis that haemophilia A carriers have reduced HR-QOL related to bleeding symptoms.	42 haemophilia A carriers and 36 control subjects = 78	Case subjects were obligated or genetically verified haemophilia A carriers age 18 to 60 years. Control subjects were mothers of children with cancer who receive care at the Vanderbilt pediatric hematology-oncology clinic.	Haemophilia A carriers had significantly lower median scores for the domains of “Pain” (73.75 versus 90; *p* = 0.02) and “General health” (75 versus 85; *p* = 0.01) compared to control subjects. Such findings highlight the need for further investigation of the effect of bleeding on HR-QOL in this population.
Goto et al., 2016 [22]	Strategies to encourage physical activity in patients with hemophilia to improve quality of life	Literature review	Discuss strategies to encourage physical activity (PA) through a behavior change approach by focusing on factors relevant to hemophilia, such as benefits and bleeding risk of PA, risk management of bleeding, PA characteristics, and difficulty with exercise adherence.	14 articles reviewed	Patients with hemophilia	For patients who find it difficult to participate in PA, it is necessary to plan individual behavior change approaches and encourage improvement in self-efficacy.
Govorov et al., 2015 [23]	Heavy menstrual bleeding and health-associated quality of life in women with von Willebrand’s disease	Cross-sectional	Investigate whether women with VWD experienced heavy menstrual bleeding (HMB) and an impaired health-associated quality of life.	30 women	Women with VWD	Of the 30 women (18–52 years) that were included in the present study, 50% suffered from HMB, although the majority received treatment for HMB. In addition, almost all the included women perceived limitations in the overall life activities due to menstruation. The health-associated quality of life for women with HMB was significantly lower (*p* < 0.10) with regards to ‘bodily pain’ compared with Swedish women of the general population.
Limperg et al., 2018 [24]	Health-related quality of life, developmental milestones, and selfesteem in young adults with bleeding disorders	Cross-sectional	Assessed HRQOL, developmental milestones, and self-esteem in Dutch young adults (YA) with bleeding disorders compared to peers.	112 YA	Ninety-five YA (18–30 years) with bleeding disorders (78 men; mean 24.7 years, SD 3.5) and 17 women (mean 25.1 years, SD 3.8)	This study demonstrates that YA men with bleeding disorders show slight impairments in total HRQOL, physical functioning, school/work functioning (PedsQL_YA), and self-esteem (RSES), in comparison to their (healthy, sex-matched) peers.
Mauser-Bunschoten et al., 2021 [25]	Managing women-specific bleeding in inherited bleeding disorders: A multidisciplinary approach	Case study	To support appropriate multidisciplinary care for WBD in haemophilia treatment centers.	Two cases	Women and girls with bleeding disorders	Multidisciplinary management is important to preserve quality of life and social participation for women from menarche onwards.
McLaughlin et al., 2017 [26]	Predictors of quality of life among adolescents and young adults with a bleeding disorder	Cross-sectional	Describe factors related to HRQoL in adolescents and young adults with hemophilia A or B or von Willebrand disease.	108 respondents	Volunteers aged 13 to 25 years with hemophilia or von Willebrand disease	Efforts should be made to prevent and manage chronic pain, which was strongly related to physical and mental HRQoL, in adolescents and young adults with hemophilia and von Willebrand disease.
Neuner et al., 2016 [27]	Health-Related Quality of Life in Children and Adolescents with Hereditary Bleeding Disorders and in Children and Adolescents with Stroke: Cross-Sectional Comparison to Siblings and Peers	Cross-sectional	Compare self-reported measures of HrQoL in a group of children and adolescents with a chronic medical condition, but no expected functional restrictions, hereditary bleeding disorder (HBD), to their siblings and peers.	144 patients	74 were patients with HBD (51.4%) and 70 were patients with stroke or TIA (48.6%)	The most relevant finding in this investigation was the overall good health-related quality of life (HrQoL)—as measured with a generic instrument, the KINDL-R questionnaire—in children with hereditary bleeding disorders

**Table 6 jcm-11-04443-t006:** Summary of articles found focused on Health Care System.

Author/Year	Title	Study Design	Objective	Number of Participants	Sample Population	Result
Arya et al., 2020 [5]	Healthcare provider perspectives on inequities in access to care for patients with inherited bleeding disorders	Cross-sectional	Understand healthcare provider perspectives regarding access to care and diagnostic delay amongst this patient population.	70 respondents	Healthcare providers	HCPs felt that there were diagnostic delays for patients with mild symptomatology (71%, N = 50), women presenting with abnormal uterine bleeding as their only or primary symptom (59%, N = 41), and patients living in rural Canada (50%, N = 35). Fewer respondents felt that factors such as socioeconomic status (46%, N = 32) or race (21%, N = 15) influenced access to care, particularly as compared to the influence of rural location (77%, N = 54).
Burke et al., 2021 [14]	Clinical, humanistic, and economic burden of severe hemophilia B in the United States: results from the CHESS US and CHESS US+ population surveys	Cross-sectional	Used the CHESS US and CHESS US+ data to analyze the clinical, humanistic, and economic burden of hemophilia B for patients treated with factor IX prophylaxis between 2017 and 2019 in the US.	44 patient records in CHESS US and 57 patient records in CHESS US+	Hemophilia B patients	Nearly all patients (85%) reported chronic pain, and the mean EQ-5D-5L utility value was 0.76 (0.24). The mean annual direct medical cost was $614,886, driven by factor IX treatment (mean annual cost, $611,971). Subgroup analyses showed mean annual costs of $397,491 and $788,491 for standard and extended half-life factor IX treatment, respectively. The mean annual non-medical direct costs and indirect costs of hemophilia B were $2371 and $6931.
Okide et al., 2020 [17]	Challenges facing community-dwelling adults with hemophilia: Implications for community-based adult education and nursing	Literature review	Discussed the challenges facing community-dwelling adults with hemophilia and the implications for community-based adult education and nursing.	Final number of articles reviewed were not disclosed	Community dwelling adults with hemophilia	Sustainable efforts are needed in the provision of local, national and international leadership and educational resources to improve and sustain health care for community-dwelling adults with hemophilia.
Pinto et al., 2018 [18]	Sociodemographic, Clinical, and Psychosocial Characteristics of People with Hemophilia in Portugal: Findings from the First National Survey	Cross-sectional	Describe sociodemographic, clinical, and psychosocial characteristics of PWH of all ages in Portugal	146 males with hemophilia	106 adults, 21 children/teenagers between 10 and 17 years, 11 children between 6 and 9 years, and 8 children between 1 and 5 years	High prevalence of joint deterioration and pain, with the ankles and the knees being the most affected joints among all age groups. A significant impact of hemophilia on professional and economic levels was particularly evident. Moreover, significant anxiety and depression symptoms were found on 36.7% and 27.2% of adults, respectively, and a belief of chronicity and symptoms unpredictability was particularly prominent among adults and children/teenagers. QoL was moderately affected among adults, but less affected in teenagers and children.

**Table 7 jcm-11-04443-t007:** Summary table of the SDOH components and respective clinical outcome.

Study Author, Year	Bleed Frequency	Chronic Pain	Mortality	Cost	Quality of Life
*Economic Stability*					
Burke et al., 2021 [14]	The mean annual bleed rate (ABR) from CHESS US was 1.73 (SD 1.39; median 2.0; Table 2). At least one bleed-related hospitalization was reported by 9.1% of CHESS US patients in the previous year, with a mean length of stay of 0.3 days.	The mean reported EQ-5D-5L score was 0.74 (SD 0.26). More than one-quarter (28%) of patients from CHESS US+ reported chronic pain ratings ≥ 6/10, and half (56%) reported pain 1–5/10 on average over the past year.		Total annual direct medical costs of hemophilia B from CHESS US were $614,886, driven almost entirely by the cost of FIX treatment ($611,971). The annual direct medical cost of hemophilia B excluding FIX treatment was $2885.	Hemophilia B is known to cause substantial functional limitations and reduced health-related quality of life (HRQoL).
Okide et al., 2020 [17]				Many community-dwelling adults with hemophilia may choose to work in jobs that are unsuitable for them, so as to obtain or maintain insurance coverage. At the same time, many with insurance coverage face rising costs of co-payments and lifetime restrictions. In addition to uncertainty about their ability to keep working, health insurance concerns can be distressing to people with hemophilia and their families.	
Pinto et al., 2018 [18]	The occurrence of bleeding episodes in the previous year was reported by 71 (67%) adults, 15 (71.4%) children /teenagers, and 6 (54.5%) and 4 (50%) young children, with the mean number of bleeding episodes varying from 4.53 (SD = 3.36) in the group of children/teenagers to 14.94 (SD = 16.90) among adults.	Pain was reported by a vast majority of participants of all age groups (>18 years: 82 [77.4%]; 10–17 years: 16 [76.2%]; 6–9 years: 9 [81.8%]; 1–5 years: 4 [50%]). Pain in the lower limbs was considered to have the greatest impact, namely, in the ankles (>18 years: 31 [37.8%]; 10–17 years: 7 [43.9%]; 6–9 years: 5 [55.5%]; 1–5 years: 1 [25%]). Pain showed a wide duration range, varying from 1 month in three age groups to 612 months (51 years) in the adults group.			A36 Hemofilia-QoL global mean score was 96.45 (SD = 27.33), with subscale scores for each specific domain also being reported. Considering CHO-KLAT, mean scores were 75.63 (SD = 12.06) for the 10 to 17 years old group and 76.32 (SD = 11.89) for the 6 to 9 years old group.
*Neighborhood and Physical Environment*					
Arya et al., 2020 [5]	With regards to which factors might affect care received by women with inherited bleeding disorders, the majority of respondents felt that lack of patient awareness around “normal” versus “abnormal bleeding” (90%, N = 63) and lack of HCP awareness (73%, N = 51) were the main barriers to care. For almost all respondents (71%, N = 49), symptoms of excessive bleeding were the most common reason for referral, followed by a positive family history (14%, N = 10) and abnormalities seen on routine bloodwork (13%, N = 9)				When asked how satisfied they thought their patients were with their quality of life, 3% of respondents (N = 2) felt that their patients were very satisfied, 58% (N = 40) felt they were satisfied, 20% (N = 14) felt they were neither satisfied nor dissatisfied, 1.5% (N = 1) felt they were dissatisfied, and none felt that they were very dissatisfied. Seventeen percent (N = 12) were uncertain. With regards to how long they estimated their patients’ travel time to clinic appointments to be, approximately half (48%, N = 34) estimated 30 minutes to one hour; 33% (N = 23) indicated one hour to two hours, 11% (N = 8) estimated over two hours, and a minority (6%, N = 4) said it took less than 30 minutes. When then asked whether they thought that access to a multidisciplinary clinic could improve quality of care for women with inherited bleeding disorders, the vast majority (90%, N = 63) indicated yes; one respondent (1.5%) indicated no, and 6 respondents (8%) were unsure.
*Education*					
Okide et al., 2020 [17]				*See health outcome summary above*	
Pinto et al., 2018 [18]	*See health outcome summary above*	*See health outcome summary above*			*See healh outcome summary above*
*Community and Social Context*					
Atiq et al., 2018 [19]	Patients who participated in sports had a higher bleeding score item for muscle haematoma 0.38 ± 0.95 vs 0.24 ± 0.75 (*p* = 0.023), which remained significant after correction for age and sex (OR = 1.26, 95% CI: 1.04-1.53).				Authors found a linear association between more hours of physical activity per week and a better general health status (*p* < 0.001).
Okide et al., 2020 [17]				*See healh outcome summary above*	
Pinto et al., 2018 [18]	*See health outcome summary above*	*See health outcome summary above*			*See healh outcome summary above*
Burke et al., 2021 [14]	*See health outcome summary above*	*See health outcome summary above*		*See health outcome summary above*	*See healh outcome summary above*
Carroll et al., 2019 [20]		Participants reporting a higher frequency of joint pain and history of joint surgery had statistically significantly lower EQ-5D-3L utility values than participants who did not experience joint pain or who had not had joint surgery previously (*p* < 0.001).			Participants with a history of long hospital stays due to hemophilia reported significantly lower SF-36 PCS scores (*p* = 0.002) than those without such a history. Also, patients with more frequent visits to medical professionals regarding hemophilia reported significantly lower SF-36 PCS scores (*p* = 0.004) than those with less frequent visits. In contrast, no significant differences were found for SF-36 MCS scores.
Gilbert et al., 2015 [21]	Haemophilia A carriers reported more severe bleeding symptoms than control subjects. The median Tosetto bleeding score of haemophilia A carriers was significantly higher than for women in the control arm (5 versus 1; *p* < 0.001). Also, haemophilia A carriers reported greater menstrual blood loss than controls as indicated by a significantly higher median pictorial blood assessment chart score (423 versus 182.5; *p* = 0.01).	Haemophilia A carriers had significantly lower scores in the “Pain” and “General Health” domains of the Rand 36-item Health Survey 1.0 than controls. Haemophilia A carriers had a median “Pain” score of 73.75 compared to a median score of 90 for control subjects (*p* = 0.02).			Our analysis indicates that haemophilia A carriers tend to have poorer HR-QOL than women who are not haemophilia A carriers, particularly in the areas of pain and general health.
Goto et al., 2016 [22]	Physical activity (PA) level has been positively correlated with bleeding risk among patients with severe and moderate hemophilia. In addition, significant differences were found in the prevalence of bleeding events, as those who exercised strenuously were more likely to incur bleeds due to trauma, and 55% of PWH actively engaged in sports reported bleeding episodes associated with PA. On the other hand, there was no significant correlation between PA level and bleeding frequency of target joints or joint function, suggesting that the risk of bleeding is dependent on bleeding history, hemostatic control, and sport participation.		Physical inactivity is the fourth leading risk factor for mortality, accounting for 6% fo deaths globally.		Continuous PA, rather than the type of exercise, is an important determinant of health-related quality of life, even for people with hemophilia.
Govorov et al., 2015 [23]	In the study population, 66.7% of the women with VWD type 1 reported HMB compared with 36.4% of the women with VWD type 2 and 25.0% of the women with VWD type 3.	In the dimension of bodily pain, the group of women with heavy menstrual bleeding (HMB) had significantly lower scores compared with those of women in the general Swedish population. This implies that the women in the study population with HMB experienced an impaired health-associated quality of life due to pain.			The health-associated quality of life according to SF-36 appeared to be lower in the study population compared with Swedish women in the general population. However, the differences in median SF-36 scores were not statistically significant.
Limperg et al., 2018 [24]					YA men with severe hemophilia (median 81.25, mean 78.86, SD 19.39) reported lower physical functioning than men with non-severe hemophilia (median 93.75, mean 93.06, SD 7.21, *p* < 0.01, r = 0.45) on the PedsQL_YA. HRQOL scores did not differ on the other PedsQL_YA scales between severity groups.
Mauser-Bunschoten et al., 2021 [25]	HMB and post-partum haemorrhages are the most frequent bleeding episodes seen associated with lower quality of life and iron deficiency anaemia.				The health-related quality of life (HRQoL) in 13 years old girls with a bleeding disorder is lower compared to their healthy peers. In contrast, there is no apparent difference in psychosocial functioning and HRQoL between young adult women with a bleeding disorder compared to peers.
McLaughlin et al., 2017 [26]		Compared with patients with no to mild chronic pain, those with moderate to severe chronic pain had 25.5-point (95% CI: 17.2, 33.8; *p* < 0.001) and 10.0-point (95% CI: 0.8, 19.2; *p* = 0.03) reductions in median PCS and MCS, respectively.			Adolescent and young adult (AYA) females with a bleeding disorder reported lower physical HRQoL when compared with AYA men in this study, even after adjustment for other sociodemographic and clinical factors.
Neuner et al., 2016 [27]					Multivariate analyses within patients, siblings, and peers revealed no differences in self-reported overall wellbeing and all KINDL-R subdimensions in group 1 (see Figure 1; all *p* > 0.05). In group 2, differences occurred inmultivariate analyses in self-reported overall wellbeing and all subdimensions (overall wellbeing *p* < 0.001, physical wellbeing *p* = 0.016, emotional wellbeing *p* = 0.007, self-worth *p* = 0.046, family relatedwellbeing *p* = 0.010, friend-related wellbeing *p* < 0.001, and school-related wellbeing *p* = 0.003).
*Health Care System*					
Arya et al., 2020 [5]	*See health outcome summary above*				*See health outcome summary above*
Burke et al., 2021 [14]	*See health outcome summary above*	*See health outcome summary above*		*See health outcome summary above*	*See health outcome summary above*
Okide et al., 2020 [17]				*See health outcome summary above*	
Pinto et al., 2018 [18]	*See health outcome summary above*	*See health outcome summary above*			*See health outcome summary above*

The greyed out cells represent the health outcomes that were not measured in each study.

## Data Availability

Not applicable.

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
