# Peer review of "Defining the Impact of Social Drivers on Health Outcomes for People with Inherited Bleeding Disorders"

_jcm, 2022, doi:10.3390/jcm11154443_

Round 1
Reviewer 1 Report
This paper reviews an important often unanswered question of the SDOH on clinical outcome, and is something important for us to improve on particularly with patients with chronic diseases like haemophilia.
I have some suggestions:
1. Introduction is too long and can be shortened.
2. Methods need to be much more succinct for the readership. Table 1 should be significantly shortened and added to appendix (or omited)
3. The results of these meta-analysis is difficult to follow and would benefit from a summary table that incorporates all of the SDOH components and respective clinical outcome.
4. Figures have to be significantly reduced
5. Conclusion - should be succinct about the findings in this study only and not a preamble regarding other aspects which the authors are interested in
Reviewer 2 Report
Comments:
The study question is important. I commend the authors for this interesting piece of work.
My main comment is that I would move the focus on the results from statistical significance to reporting effect sizes and the relative confidence intervals, and helping the reader with the interpretation. E.g.: in one study (ref) economic instability was associated with worse quality of life (risk difference …, 95% CI … ). Moreover, it would be desirable to formally assess the risk for bias with a validated instrument (e.g., ROBIS I). This should then be reflected in the discussion.
Other comments as follows:
Materials and methods:
1. Eligibility criteria:
a. “Published in English between 2011 and 2021”
Please justify the time frame. The lack of a MeSh term for SDOH before 2014 can be overcome by also using free text in the search strategy, as the authors already appropriately did.
b. “Available in free full text”
This is an unusual exclusion criterium. How many references have been excluded for this reason? One of you have a university affiliation. Maybe the university provides access to at least some of the pay-walled content?
c. Did you include all types of study design, setting, and FUP duration? Please specify.
d. Regarding point c above, you also included a literature review. This is unusual. If you are surveying primary literature, you might want to use the review to find he references included, and eventually screen those for eligibility in your review, as opposed to including the review itself.
2. Search strategy: I think this does not affect the results of the search, but there might be some inaccuracy in combining the searches. My guess is that you want to combine #1 (PWiBD) AND SODH (2-11, combined with OR). If this is the case:
a. #12 is probably not needed
b. #13 should not include #1
3. Reporting the outcome definition among the study characteristics would be helpful for interpretation.
Results:
4. “Overall, three were categorized as Economic Stability, one was categorized as Neighborhood and Physical Environment, 0 were categorized as Food, two were categorized as Education, twelve were categorized as Community and Social Context, and four were categorized as Health Care Systems (Table 7).”
You might want to introduce these categories in the methods section first, to help the reader.
Conclusions:
5. Limitations:
a. Only in English, only recent, only pubmed, screening not in duplicate
Minor comments:
6. Using a PRISMA flow diagram, the search resulted in 1,466 screened articles. After conducting a title review for inheritable bleeding disorder populations, 27 articles were identified for retrieval.
The PRISMA flow diagram was used for reporting, so I would rephrase as follows:
“After duplicates removal, the search resulted in 1,466 screened articles. After conducting a title review for inheritable bleeding disorder populations, 27 articles were identified for retrieval (see the PRISMA flow diagram, Figure 4).”
Round 2
Reviewer 1 Report
No further comments